# Fh15 Reduces Colonic Inflammation and Leukocyte Infiltration in a Dextran Sulfate Sodium-Induced Ulcerative Colitis Mouse Model

**DOI:** 10.3390/cells14110799

**Published:** 2025-05-29

**Authors:** María Del Mar Figueroa-Gispert, Claudia M. Ramos-Lugo, Carlimar Ocasio-Malavé, Rizaldy P. Scott, Jared T. Ahrendsen, Mercedes Gomez-Samblas, Antonio Osuna, Stephanie M. Dorta-Estremera, Ana M. Espino

**Affiliations:** 1Department of Microbiology and Medical Zoology, University of Puerto Rico-Medical Sciences Campus, San Juan, PR 00936, USA; maria.figueroa14@upr.edu (M.D.M.F.-G.); claudia.ramos11@upr.edu (C.M.R.-L.); carlimar.ocasio@upr.edu (C.O.-M.); stephanie.dorta@upr.edu (S.M.D.-E.); 2Mouse Histology & Phenotyping Laboratory, Feinberg School of Medicine, Northwestern University, Chicago, IL 60611, USA; rizaldy.scott@northwestern.edu (R.P.S.); jared.ahrendsen@northwestern.edu (J.T.A.); 3Institute of Biotechnology, Department of Parasitology, University of Granada, 18071 Granada, Spain; msambla@ugr.es (M.G.-S.); aosuna@ugr.es (A.O.)

**Keywords:** ulcerative colitis, *Fasciola hepatica*, fatty acid binding protein, myeloperoxidase, chitinase-3 like-protein-1, S100A9, TNF-α, IL-1β, neutrophils, macrophages

## Abstract

Ulcerative colitis (UC) is the most prevalent inflammatory bowel disease (IBD) in the USA. Current treatments present clinical limitations, underscoring the need for innovative therapeutics that promote an anti-inflammatory immune response. This study evaluates the anti-inflammatory potential of Fh15, a recombinant *Fasciola hepatica* fatty acid binding protein, in a DSS-induced UC mouse model. Our results demonstrated that Fh15 treatment significantly ameliorated the severity of colitis by reducing the disease activity index (DAI) and histopathological scores. Moreover, Fh15 also decreased the serum levels of myeloperoxidase (MPO) and chitinase-3-like protein 1 (CHI3L1), and the expression of S100A9, a calcium and zinc binding protein, which is an important marker for the pathogenesis of UC. Furthermore, Fh15 downregulated pro-inflammatory cytokines TNFα and IL-1β in the distal colon, suggesting modulation of macrophage activity. Immunohistochemistry analysis revealed significantly reduced neutrophil and macrophage infiltration in UC Fh15-treated mice. These findings highlight the therapeutic potential of Fh15 for UC, as it modulates inflammatory responses, reduces leukocyte infiltration, and preserves colon integrity.

## 1. Introduction

Ulcerative colitis (UC) is the most common inflammatory bowel disease (IBD) affecting approximately 286 individuals for every 100,000 persons per year in the USA [1]. Its high prevalence also extends to Canada and many countries in Western Europe. Although the underlying etiology of UC remains unclear, it is thought to result from a combination of factors, including genetic predisposition, immunoregulatory issues, environmental influences, and gut microbiota [2]. These factors contribute to a chronic inflammatory state driven by colitogenic, pro-inflammatory T helper type-1 (Th1) cells, characterized by the production of large amounts of interferon-gamma (IFN-γ) and tumor necrosis factor (TNF) [3]. There is no cure for UC, and existing treatments have significant limitations that prevent many patients from achieving remission [4,5]. Therefore, there is a need to incorporate new treatments for UC.

Previous epidemiological studies and clinical trials have suggested that helminth infections might protect people from IBDs [6]. Helminths have co-evolved with humans for millions of years and exert a strong immunomodulatory effect on their host. They polarize the immune response toward an anti-inflammatory Th2/T-regulatory response, which can only be achieved by suppressing the Th1-inflammatory response [7]. This strategy leads helminths to establish a prolonged chronic infection in their mammalian host [8]. The prevalence of helminth infections contrasts with the patterns observed for IBDs and other autoimmune diseases. Helminth infections are endemic in underdeveloped countries but are rarely reported in Western countries. Interestingly, the noticeable rise in IBD cases in developing countries has been linked, among other factors, to the widespread application of intensive anti-parasitic treatments aimed at deworming the population. It is believed that such a deworming campaign might have disrupted the long-term regulatory network established by helminths and may be in part responsible for the emergence of immunological disorders [9].

Considering that T-cells are critical for IBD and helminth infections, the reciprocal cross-regulation between Th1 and Th2 cells suggests that the polarization of a Th2 response by helminths could prevent or minimize the effects of Th1-mediated diseases. It has been demonstrated that the infection of mice with the nematode *Trichinella spiralis* ameliorates subsequent induced colitis, which is associated with a downregulation of the Th1 response [10]. Similarly, infection with *Fasciola hepatica*, one of the most widely distributed trematodes worldwide, has been shown to attenuate the clinical symptoms of murine autoimmune encephalomyelitis [11] and prevent type-1 diabetes development in non-obese diabetic mouse models [12]. Experimental infections with schistosome cercariae have been shown to confer protection against type-1 diabetes [13], encephalomyelitis [14], and colitis [15]. Moreover, clinical trials have provided evidence that the therapy with ova from the nematode *Trichuris suis* is effective for treating Crohn’s disease and UC without any adverse effects [16]. However, the use of ova, cysts, or deliberate exposure to helminth larvae as a therapeutic approach may be unappealing to patients and ethically unacceptable to many. The inability to control the progression of such helminthic infections remains a major concern. Additionally, the strong polarizing immune response toward Th2 induced by these parasites is highly nonspecific, potentially rendering the host incapable of responding effectively to concurrent infections that require a Th1-mediated immune response for protection [12,17]. In this sense, the identification of helminth-derived molecules that ultimately mediate host immune modulation could be a more attractive and feasible therapeutic option [18,19].

Our research group has identified Fh15, the recombinant form of a member of the *F. hepatica* fatty acid binding protein family and demonstrated that it exhibits powerful anti-inflammatory properties. A single dose of Fh15 administered intraperitoneally to mice exposed to lethal doses of lipopolysaccharide (LPS) *E. coli* can significantly suppress the cytokine storm [20] and increase the survival rate [21]. When Fh15 was administered prophylactically via intravenous infusion to non-human primates challenged with live *E. coli*, it was shown to suppress the cytokine storm, bacteremia, endotoxemia, and other inflammatory markers associated with sepsis [22]. Although sepsis and ulcerative colitis are distinct diseases, they share common characteristics: both can be caused by bacterial infections, trigger excessive inflammatory responses, and result in severe health complications. In this study, we investigated the potential of Fh15 as a therapeutic agent to suppress intestinal inflammation caused by ulcerative colitis using a DSS-induced colitis model in C57BL/6 mice. Our results demonstrate that Fh15 can mitigate intestinal inflammation by modulating the colonic immune cell infiltration and suppressing pro-inflammatory markers associated with the severity of UC pathology. Thus, Fh15 holds promise for further research aimed at developing a new helminth-derived treatment against ulcerative colitis.

## 2. Materials and Methods

### 2.1. Animals and Ethics Statement

A total of 50 C57BL/6 male mice (6–8-week-old) were obtained from Charles River Laboratories and maintained under standard conditions at 21 °C and with a 12 h light–dark cycle, with access to food and water ad libitum. All procedures were conducted according to the regulations of the Ethics Institutional Animal Care and Use Committee of the University of Puerto Rico-Medical Sciences Campus (Protocol No. 7870123). Euthanasia was performed under deep anesthesia.

### 2.2. Recombinant F. hepatica FABP (Fh15)

The recombinant Fh15 used in the present study was obtained as previously described [22] and consisted of 10 mL of purified Fh15 (2.29 mg/mL, <0.4 EU/mg) with >90% purity confirmed by LC-MS. Fh15 was expressed in *Bacillus subtilis* as a fusion protein with a 6His at the amino terminus and purified by a Ni^+^-agarose column. The aliquots of purified Fh15 used in the present study remained stored at −80 °C until use.

### 2.3. Dextran Sulfate Sodium (DSS) Colitis Induction and Fh15 Treatment

First, 5 animals were randomly assigned into 1 of 6 groups (PBS, DSS, Fh15, DSS-Fh15 50 µg, DSS-Fh15 100 µg, and DSS-Fh15 150 µg) to perform a dose-response experiment. The healthy control group, named PBS, drank normal water, and received three intraperitoneal (i.p.) injections on days 1, 3, and 5 of 50 µL of 0.1 M phosphate-buffered saline (PBS) at pH 7.2, free from endotoxin. The group designated as Fh15 drank normal water and received three i.p. injections of 50 µg of Fh15 diluted in 50 µL of PBS on days 1, 3, and 5. Acute colitis was induced in groups labeled as DSS and DSS-Fh15 by providing ad libitum access to autoclaved drinking water with 4% (*w*/*v*) dextran sulfate sodium (DSS; 40 kDa, Sigma-Aldrich, Burlington, MA, USA) for 7 days as described by Chassaing et al. (2014) [23]. The DSS-Fh15 groups received three i.p. doses of 50 µg (2.0 mg/kg), 100 µg (4.0 mg/kg), or 150 µg (5.0 mg/kg) of Fh15 according to their respective group. These doses were administered on days 1, 3, and 5 of DSS-water consumption (Figure 1). The number of Fh15 injections was determined based on studies reported in the literature using other helminth antigens [18]. At the end of the experimental period (Day 7), animals were anesthetized and bled via the orbital vein using capillary tubes. The serum was obtained by blood centrifugation (10,000× *g*, 4 °C, 10 min). After bleeding, animals were euthanized and necropsied to collect their colon and spleen.

### 2.4. Disease Activity Index (DAI)

The disease activity index (DAI) was used as the primary criterion to assess colitis severity, as described by Wang et al. (2017) [19]. Mice were monitored daily for changes in body weight, stool consistency, and the presence of blood in the stool. Disease severity was assessed by a clinical scoring system as follows: weight loss, 0 (<2%), 1 (≥2%–<5%), 2 (≥5%–<10%), 3 (≥10%–<15%), or 4 (>15%); stool consistency, 0 (normal), 1 (softer stool), 2 (moderate diarrhea), or 3 (diarrhea); and the presence of blood in stool, 0 (no rectal bleeding), 1 (positive Hemoccult/no visible blood), 2 (visible blood in stool), or 3 (fresh rectal bleeding). The summed scores for weight loss, stool consistency, and the presence of blood in stool were used as the DAI.

### 2.5. Macroscopic Score and Histopathological Scoring

The colons removed from each animal on the euthanasia day were macroscopically examined by using a modified established scoring system [24] to indirectly measure inflammation. We measured colon length in centimeters (cm) from the cecum to the rectum, adhesion, inflamed length (cm), and bowel thickness (mm), and observed the presence of hemorrhage, fecal blood, and diarrhea on the euthanasia day (Table 1). All scores per mouse were summed to determine the final macroscopic score. To allow easy handling, the colons were divided into several segments and some portions were slit open longitudinally and carefully cleaned up to remove the content. Next, using wooden stick each segment was rolled up longitudinally with the mucosa outwards. Finally, the preparations were fixed for 48 h in 10% formalin. Tissues were embedded in paraffin and cut into 4 µm thick sections on positively charged slides. Dewaxed sections were routinely stained with hematoxylin–eosin (H&E) dyes for histological evaluation [25]. The histopathological score was calculated according to well-accepted parameters described elsewhere [26] in a blinded fashion by an expert pathologist (J.T.A.). The evaluation parameters were based on the criteria established by Sann, H. et al. (2013) [26], which assess the extent of inflammation, leukocyte infiltration, crypt damage, crypt abscesses, submucosal edema, goblet cell loss, and reactive epithelial hyperplasia. These factors were evaluated using a scoring system ranging from 0 to 4. All scores per parameter and per mouse were summed to determine the final histopathological score.

### 2.6. Serum Myeloperoxidase and Chitinase-3 Like-Protein-1 Concentration

To measure the levels of myeloperoxidase (MPO) in serum, an enzyme released primarily by neutrophils [27], we utilized a Mouse Myeloperoxidase SimpleStep ELISA^®^ Kit (Abcam, Cambridge, UK; ab275109). The chitinase-3-like protein 1 (CHI3L1) levels in serum were measured using a Mouse YKL-40/CHI3L1 ELISA^®^ kit (Abcam; ab238262). CHI3L1 is secreted by different cell types and is considered an important marker of inflammation [28]. Absorbance for both MPO and CHI3L1 was measured spectrophotometrically at 450 nm, with concentrations reported in pg/mL.

### 2.7. Colonic Cytokines Gene Expression

TNF-α and IL-1β expression were measured in the distal colon sections of each mouse. Tissue was stored in TRIzol™ reagent and kept at −80 °C until use. Total RNA was extracted using the AllPrep^®^ DNA/RNA/Protein kit (Qiagen, Germany, Cat. No. 80004). As DSS can inhibit the qRT-PCR by altering the binding between the reverse transcriptase and primed RNA [29], we removed DSS by purifying the mRNA from total RNA using the RNeasy^®^ Pure mRNA Bead kit (Qiagen, Germany, Cat. No. 180244). Purified mRNA (10 ng) was used to generate cDNA using the Applied Biosystems Power SYBR™ Green RNA-to Ct™ 1 Step kit (Thermo Fisher Scientific, Waltham, MA, USA, Cat. No. 4389986). The cDNA was subjected to real-time quantitative PCR using the QuantStudio™ 3 system (Thermo Fisher Scientific, Waltham, MA, USA) with the QuantStudio™ Design and Analysis software (v1.5.2). The cytokine primer sequences by Sigma Aldrich are shown in Table 2. The relative expression of each gene was calculated by 2−ΔΔCt (RQ) using GAPDH as an internal reference.

### 2.8. Colon Immune Cells Infiltration and Calcium Binding Protein Marker

Dewaxed colon sections were subjected to standard immunohistochemistry (IHC) procedures to determine immune cell infiltration and measure the levels of S100 calcium binding A9 protein (S100A9), which is a protein secreted by granulocytes and is highly elevated in patients with UC [30]. The primary antibodies used were rabbit anti-mouse CD3ε (T-cells), anti-F4/80 (macrophage), anti-Ly6G (neutrophil), and anti-S100A9 (Table 3). As the secondary antibody, we used an anti-rabbit antibody-horseradish peroxidase (HRP) polymer conjugate (MACH2, Cat# RHRP520MM, Biocare, Pacheco, CA, USA) and as the chromogenic substrate, we used 3,3′-diaminobenzidine (DAB) (Cat. No. SK-4103-400, Vector Labs., Newark, NJ, USA). IHC slides were counterstained with hematoxylin. Brightfield images were captured with an inverted confocal microscope equipped with a 40× objective. The percentage of antibody-positive tissue areas was evaluated using Nikon NIS Elements AR Software v.5.20 [31]. First, a binary layer threshold was created to exclude unstained colonic tissue from the antibody-stained tissue area. Then, three regions of interest (ROIs) were created, each with a total area of 5057.61 µm^2^ and randomly distributed across the tissue area. This process allows the quantification of the ROI’s area fraction occupied by antibody staining. A total of three images from different colon regions, per animal, were analyzed for each antibody of interest. The percentage area of the thresholder signal was exported and normalized to the average value of non-treated tissues. Representative immunostaining images were adjusted using Nikon NIS-Elements AR Software version 5.20 with the following settings: high contrast set to 450, low contrast to 20, and saturation to 10%.

### 2.9. Splenic T Cell and Leucocyte Populations Profiling

Spleens removed from each animal on euthanasia day were cut into 3 to 5 mm pieces and smashed using a syringe plunger and then filtered through a 70 μm cell strainer. Red blood cells were removed using red blood lysing buffer (Sigma-Aldrich, Burlington, MA, USA). Prepared single-cell suspensions of 5 × 10^5^ cells per mouse were stained using a 10-color antibody panel for multiparametric flow cytometric analysis. Cells were incubated with an antibody cocktail containing Fc-block (1:500) and live/dead Aqua Zoombie (1:400), and then stained with antibodies against CD3 BV711 (1:400), CD4 PE-Dazzle 594 (1:200), CD8 FITC (1:200), CD11b BV650 (1:400), CD11c Pacific Blue (1:200), and CD86 PerCP/Cy5.5 (1:200) (all from BD Biosciences, San Jose, CA, USA). Staining was performed for 30 min at 4 °C in the dark, followed by washing with FACS buffer and fixation with fix buffer (BD Biosciences, Franklin Lakes, NJ, USA) for 10 min at 4 °C. After fixation, cells were washed, resuspended in FACS buffer, and stored at 4 °C until acquisition. Data were acquired using a 2-laser BD FACS Celesta flow cytometer and analyzed with FlowJo software version 10.6.2 (FlowJo LLC, Ashland, OR, USA). The gating strategy is shown in Appendix A.

### 2.10. Statistical Analysis

Results are expressed as means ± SE values. Statistical significance was determined by one-way or two-way ANOVA with Dunnett’s multiple comparisons using GraphPad Prism software v.8 (GraphPad Software Inc., La Jolla, CA, USA). Data with *p* < 0.05 were considered statistically significant.

## 3. Results

### 3.1. Fh15 Treatment Reduces Disease Activity Index in DSS-Induced UC Mice

We used C57BL/6 mice, the prototypical Th1-type mouse strain [32], to develop the DSS-induced UC mouse model. To optimize the Fh15 treatment dose, we first accomplished a dose–response experiment in which three different DSS groups received i.p. injections with 50 μg, 100 μg, or 150 μg of Fh15 on days 1, 3, and 5. Each Fh15 administration corresponded to doses of 2.0, 4.0, or 5.0 mg/kg body weight, respectively. Animals were monitored daily, and the disease activity index (DAI) was calculated based on weight loss, stool consistency, and presence of blood in stool. Compared with the baseline, mice in the DSS group exhibited a significant weight loss, with mice showing a mean loss of 19.39% by day 7 (Figure 2A, Appendix A). The DSS group also displayed the highest scores for rectal bleeding and diarrhea, achieving a maximum disease activity index of 10 (Figure 2B and Appendix A), indicating clear and progressive intestinal damage. In contrast, animals that received the treatment with Fh15, irrespective of the dose assayed, showed significant reductions in DAI (*p* < 0.0001) from day 2 to 7 (Appendix A). Since no differences in DAI reductions were observed among the Fh15 doses, it was not possible to determine the optimal dose at that time. However, it is important to mention that DSS mice treated with 50 μg of Fh15 experienced a *significant* reduction in weight loss, averaging 10.62% compared with untreated colitis mice on day 7 (Figure 2A, Appendix A). These findings indicate that Fh15 effectively alleviates clinical scores associated with DSS-induced ulcerative colitis.

### 3.2. Fh15 Significantly Prevents Colon Shortening and Decreases Macroscopic Score in DSS-Induced UC Mice

Since mice with acute colitis tended to exhibit colon shortening, we measured the colon length, in centimeters (cm), as an additional criterion to assess the therapeutic effect of Fh15. Consistent with the DAI results, a significant colonic shortening was observed for the DSS group (Figure 3A). For the DSS groups treated with Fh15 at different concentrations, the results indicated that a dose of 2.0 mg/kg (DSS-Fh15 50 μg group) of Fh15 was the only one to show a significant effect compared with the DSS group (*p* < 0.001). In contrast, the groups of 100 μg and 150 μg demonstrated little to no impact on colon shortening (Appendix A). Although all Fh15 doses significantly reduced DAI and the macroscopic score (Appendix A), the dose of 2.0 mg/kg body weight was selected for subsequent experiments based on its effect on colon shortening. We also measured adhesion, inflamed length (cm), and bowel thickness (mm), and observed the presence of hemorrhage, fecal blood, and diarrhea on day 7 to calculate the macroscopic damage with a modified established scoring system (Table 1) [24]. The higher score for the DSS group was 11.10 (Appendix A). In comparison, the mean colon macroscopic scores for DSS-Fh15-treated mice were significantly lower, with a minimum score of 5.90, indicating substantial suppression (Figure 3B, Appendix A). These data indicate that Fh15 reduces colon shortening and lowers the macroscopic score, decreasing UC inflammation.

### 3.3. Fh15 Ameliorates Histological Alterations in DSS-Induced UC Mice

To evaluate if Fh15 was attenuating histological alterations induced by DSS administration, colonic hematoxylin and eosin staining was used and double-blinded scored by a pathologist. Consistent with the results described above, colonic histological alterations were observed in the DSS group, which had the highest scores for each assessment criterion (Appendix A). DSS-Fh15 mice exhibited a lower histopathological score compared with the DSS group, suggesting a potential advantageous effect of the treatment. As shown, DSS-Fh15 displayed a lower extent of inflammation, mucosa with a better goblet cell architecture, and less crypt distortion (Figure 3C, Appendix A). Therefore, Fh15 treatment helps to reduce structural changes associated with ulcerative colitis in the colons of mice, helping to preserve intestinal crypts and epithelial cells.

### 3.4. Fh15 Decreases Serum Levels of Myeloperoxidase and CHI3L1 While Suppressing S100A9 and Pro-Inflammatory Cytokines in Colonic Tissues of DSS-Induced UC Mice

Consistent with the anti-inflammatory effects showed by Fh15, the levels of MPO (*p* < 0.05) and CHI3L1 (*p* < 0.01) were significantly reduced in UC mice when treated with Fh15 (Figure 4A,B). Next, we determined the distal colon expression of two key pro-inflammatory cytokines, TNFα and IL-1β, which were found to be overexpressed in untreated DSS mice (Figure 4C,D). Importantly, the expression of TNFα and IL-1β were found significantly reduced in the DSS-Fh15 group (*p* < 0.0001 and *p* < 0.01, respectively). Moreover, IHC analysis revealed a significant increase in S100A9 levels in the colons of untreated DSS mice, which were found to be significantly decreased (*p* < 0.01) in DSS-Fh15 mice (Figure 4E,F). Healthy animals that were treated with Fh15 showed background levels (similar to PBS) of MPO, CHI3L1, S100A9, TNFα, and IL-1β, which indicates that Fh15 alone does not contribute to the inflammation or overexpression of any inflammatory marker (Figure 4). These results indicate that Fh15 reduces important pro-inflammatory markers in DSS mice, indicating its potential to mitigate intestinal inflammation associated with ulcerative colitis.

### 3.5. Fh15 Modulates Colonic Tissue Leukocyte Cell Infiltration of DSS-Induced UC Mice

To evaluate the effect of Fh15 on intestinal immune cell infiltration, we performed IHC analysis using primary antibodies specific for macrophages (F4/80+), neutrophils (Ly6G+), and T-cells (CD3+). As shown in our results, the mean percentage of F4/80-positive (Figure 5A) and Ly6G-positive (Figure 5B) areas was significantly reduced in animals that received Fh15 treatment (*p* < 0.05 and *p* < 0.01, respectively). In contrast, no significant differences were observed in the mean percentage of CD3⁺ areas between the PBS, DSS, Fh15, and DSS-Fh15 groups (Figure 5C). In line with this observation, colonic tissue images from DSS-treated mice showed elevated levels of F4/80⁺ and Ly6G⁺ markers, indicating increased macrophage and neutrophil infiltration. However, immunohistochemical analysis for T cells using anti-CD3+ primary antibody showed no statistically significant differences across the groups (Figure 5F). These findings suggest that Fh15 treatment effectively reduces macrophage and neutrophil infiltration in the colonic tissue of DSS-induced ulcerative colitis mice, and that by day 7 of 4% DSS administration, T cell responses are not yet fully established for this model. To confirm these observations, spleen lymphocytes were labeled with a panel of specific antibodies for T-cells (CD3+, CD4+, and CD8+) and then analyzed by flow cytometry. These results corroborate the IHC findings, as no significant differences were observed in T cell populations between healthy mice, DSS-induced colitis mice, or those treated with Fh15 (Appendix A). Additionally, dendritic cells (DCs) were labeled using CD11b, CD11c, and CD86 antibodies. No significant differences were observed between the DSS and DSS-Fh15 groups (Appendix A). However, a leukocyte population classified as CD11b+ CD11c- CD86+ was found to be significantly increased in DSS-colitis mice compared with healthy controls (*p* = 0.0019). This leukocyte population was significantly suppressed by Fh15-treatment (*p* = 0.0161) (Figure 6), which suggests that Fh15 suppresses the capacity of these cells to activate T cells. Appendix A present representative immunohistochemistry images of the key markers analyzed, S100A9, F4/80, Ly6G, and CD3, captured with two different objectives (10× and 40×).

## 4. Discussion

*Fasciola hepatica* is recognized as the “master of immune regulation” [33]. From the very early stages of infection, this parasite secretes myriads of molecules termed excretory–secretory products (ESPs) that are responsible for inducing major Th2 responses with concurrent suppression of Th1 responses [17,34]. It is therefore unsurprising that *F. hepatica* infection [11], its excretory–secretory products [12], or extracellular vesicles [18], have been utilized to prevent or relieve autoimmune diseases, including ulcerative colitis (UC). Our research group has characterized members of the *F. hepatica* fatty acid binding proteins (FABPs), particularly Fh15, as some of the most immunomodulatory molecules with potential biotherapeutic applications. In various experimental models of sepsis, including non-human primates, the treatment with Fh15 was able to suppress inflammatory markers associated with Th1 responses without apparent side effects and excellent tolerability [22]. Therefore, Fh15 represents one of the best characterized helminth-derived molecules, ready for use in clinical trials, owning to its immune-modulating properties. These properties led us to investigate the effects of Fh15 treatment on experimentally induced UC.

In this study, colitis was induced by providing C57BL/6 mice with ad libitum access to drinking water containing 4% DSS, a colitogenic chemical with anticoagulant properties. This method of DSS-induced colitis in mice closely resembles human UC [35]. The mechanism by which DSS induces intestinal inflammation is unclear, but it is likely to result in colonic epithelial monolayer lining damage allowing the dissemination of proinflammatory intestinal contents (e.g., bacteria and their products) into underlying tissue. Our data show that the administration of 2.0 mg/kg Fh15 to colitis-mice three times a week significantly attenuated the disease activity index (DAI), macroscopical, and histological features of acute colonic inflammation. Our results are consistent with those reported by Roig et al., 2018 [18] using extracellular vesicles from *F. hepatica* (FhEV) and were also similar to those obtained by others that used defined recombinant antigens from *Schistosoma mansoni* (GST28) [36], *Trichinella spiralis* (TsSp) [37], and *Ascaris lumbricoides* (Al-CPI) [38] in colitis mouse models. However, in our study, Fh15 was administered intraperitoneally (i.p.) concurrently with colitis induction, whereas in other studies, treatments with GST28, TsSp, or Al-CPI were administered prophylactically via subcutaneous or intraperitoneal (i.p.) routes several days or weeks before colitis induction. These observations suggest that, regardless of the type of antigen or the route of administration, helminth-antigen therapy can be effective in both prophylactic and therapeutic applications, alleviating the pathological effects of colitis thus, contributing to enhancing the regenerative capacity of the intestinal epithelium.

Because macrophages and neutrophils play essential but different roles in ulcerative colitis, we proceeded to examine the effect of Fh15 on these cells. Macrophages are particularly abundant in the gastrointestinal mucosa, especially in the lamina propria near the epithelium [39], and can promote inflammation depending on their phenotype. Macrophages can be classically activated (M1 macrophages) or alternatively activated (M2 macrophages), with M1 being the pro-inflammatory phenotype and M2 being the anti-inflammatory phenotype [40]. M1 macrophages act as the first signal to prime T cells for activation and differentiation into effector T cells, particularly CD4+ T cells. In the inflammatory stage of UC, excessive macrophage infiltration can aggravate inflammation. Although characterizing the macrophage phenotype was beyond the scope of our study, it can be assumed that intestinal macrophages in the DSS-colitis group are polarized toward the M1 phenotype, similar to the predominance of M1-type macrophages observed in the colons of patients with UC [41]. M1-type macrophages contribute to inflammation by secreting pro-inflammatory cytokines such as TNFα, which plays a crucial role in immune cell expansion by enhancing NF-κB activation [42]. Additionally, TNFα increases intestinal permeability, allowing the passage of antigens and toxic substances, which leads to intestinal inflammation [43]. IL-1β, a pro-inflammatory cytokine secreted primarily by macrophages, was also found to be overexpressed in DSS-colitis mice. The excessive production of IL-1β contributes to intestinal epithelial barrier disruption [44] and promotes Th17 cell differentiation [45], a key factor in IBD inflammation. Therefore, the observation that treatment with Fh15 significantly reduced the expression of TNFα and IL-1β in colonic tissue suggests that Fh15 might have a role in modulating the number of colonic tissue immune cells. This assumption is consistent with our previous studies in the sepsis model, which demonstrated that Fh15 increases the population of large peritoneal macrophages (LPMs), which are essentially anti-inflammatory, to perpetuate the steady state or homeostasis in the setting of an inflammatory stimulus [20].

Neutrophils are the largest population of myeloid leukocytes, and the first to arrive immediately at the infection site, providing a “first line” of defense against pathogens [46]. They contain a huge number of antimicrobial granules that allow them to destroy pathogens during phagocytosis or outside the cells [46]. Under physiological conditions, neutrophils are absent from healthy intestinal mucosa. However, at the onset of the intestinal inflammation, neutrophils are rapidly recruited from circulation. Macrophages play a critical role in the recruitment of neutrophils during UC by secreting IL-1β, and TNF-α among other chemotactic signals [47]. Neutrophils can destroy pathogens outside cells by releasing neutrophil extracellular traps (NETs) through a process termed NETosis [48]. NETs consist of modified chromatin “decorated” with bactericidal proteins such as myeloperoxidase, elastase, and histones [27]. NET components are indiscriminately cytotoxic and pro-inflammatory [49], and when released in excess, they activate and exacerbate a wide range of pathologies, including ulcerative colitis [50]. Activated neutrophils can also express S100A9 [51], a protein that makes up almost half of the intracellular protein content. S100A9 is a calcium-binding protein that is constitutively expressed by neutrophils, dendritic cells, and monocytes. S100A9 is released extracellularly under inflammatory conditions and is thought to act as a damage-associated molecular pattern (DAMP) [52]. The observation that Fh15 treatment significantly reduced the levels of CHI3L1 and S100A9 is, therefore, consistent with the substantial reduction in the number of neutrophils and macrophages in the colonic tissue of colitis mice. On the other hand, the observation that Fh15 can also suppress the levels of MPO in serum could also suggest that Fh15 may have an impact on NET production, an effect that currently is being investigated.

Interestingly, although CD4⁺ T cells are known to become overactivated during colitis and contribute to gut inflammation through the release of pro-inflammatory cytokines [53,54], studies have shown that T cells are not essential for colitis induction in the DSS-model [55,56]. This may explain the limited presence of T-cell populations by day 7 observed in this study in DSS-treated animals. However, although they are not required, some T cells still infiltrate the colon and appear to play a role in the chronic stage of colitis [57,58]. Previous studies have shown that, during the acute phase of DSS-induced colitis, the colon is predominantly infiltrated by innate immune cells, whereas T cell infiltration becomes more pronounced during the late acute or chronic phases [59], often peaking around day 12 after the resumption of regular water intake [60]. Moreover, the observation that a CD11b+ CD11c- population increased the expression of CD86+ in the DSS-colitis mice and was suppressed by the Fh15-treatment strongly suggests that Fh15 suppresses the activation of myeloid cells. Since the CD11b+ CD11c- cell population excludes the presence of dendritic cells, we could assume that the spleen leukocyte population suppressed by Fh15 could mostly be comprised of monocytes, macrophages, neutrophils, and natural killer cells, which express CD11b+. Importantly, CD86 is a costimulatory molecule highly expressed in antigen-presenting cells and is essential for providing the secondary signals required for T cell activation [61,62,63]. Therefore, it is possible to speculate that Fh15 may suppress CD86-expressing leukocyte populations during the acute phase of colitis as a mechanism to prevent excessive T cell activation in the later stages of colitis.

Additionally, since Fh12, a native fatty acid binding protein of *F. hepatica*, has been proposed as a TLR4 antagonist [64] and we have demonstrated that Fh12 and Fh15 share immunologic and antigenic properties [65], we could speculate that Fh15 could also suppress the overexpression of TLR4, which is directly associated with the severity and progression of DSS-induced colitis [66,67]. Given that S100A9 functions as a ligand for TLR4 [68], and that TLR4 and CHI3L1 signaling pathways are interconnected in inflammatory bowel diseases [69], the ability of Fh15 to suppress both S100A9 and CHI3L1 suggests that it may also modulate additional molecules involved in these associated signaling pathways. Concurrently, Fh15 could upregulate cytokine repair factors like PGE2 and GM-CSF, which are also associated with TLR4-signaling [70], or it can increase M2-type population, which plays an essential role in intestinal epithelium repair [71]. Moreover, as a helminth-derived molecule, Fh15 may also suppress colonic inflammation by promoting the expansion of regulatory T cells, which play a critical role in maintaining gut immune homeostasis [72]. Further studies are being designed to investigate the possible role of Fh15 in all these putative mechanisms of action.

## 5. Conclusions

In conclusion, this study demonstrates that Fh15, a recombinant *Fasciola hepatica* fatty acid binding protein, significantly reduces inflammation in a DSS-induced ulcerative colitis mouse model. Fh15 administration mitigated clinical symptoms, decreased pro-inflammatory markers, reduced macrophage and neutrophil infiltration in the colon, and decreased the activation of CD11b+ CD11c- spleen leucocytes, contributing to the restoration of intestinal mucosal integrity and reduction of inflammatory damage. These findings highlight the therapeutic potential of Fh15 for modulating the immune response and treating ulcerative colitis, making it a promising candidate for drug development targeting inflammatory bowel diseases.

## Figures and Tables

**Figure 1 cells-14-00799-f001:**
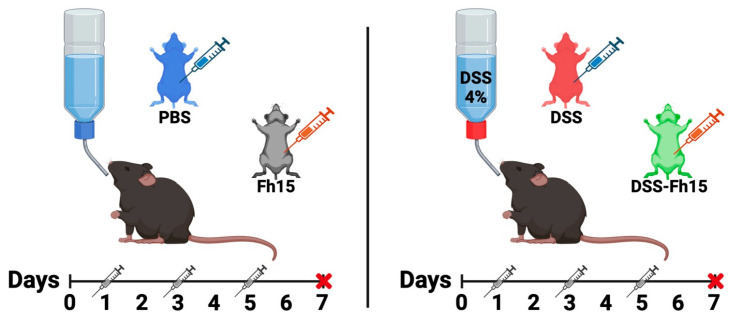
DSS-induced ulcerative colitis and Fh15 treatment administration in C57BL/6 male mice. Colitis was induced in C57BL/6 male mice by providing 4% (*w*/*v*) dextran sulfate sodium (DSS) in autoclaved drinking water for 7 days to the DSS and DSS-Fh15 groups. PBS and Fh15 groups received normal drinking water. The Fh15 group received three intraperitoneal (i.p.) injections of Fh15 (2 mg/kg body weight). The DSS-Fh15 group was divided into three subgroups that received different i.p. doses of Fh15: 50 µg, 100 µg, and 150 µg, corresponding to 2.0 mg/kg, 4.0 mg/kg, and 5.0 mg/kg body weight, respectively. Control groups (PBS and DSS) received an equivalent volume of endotoxin-free PBS. Fh15 or PBS was administered on days 1, 3, and 5. Red cross: euthanasia day. Blue syringe: PBS administration. Orange syringe: Fh15 administration. Gray syringe: Day of administration. Created in Created in BioRender. Figueroa, M. (2025) https://BioRender.com/n0aacjy.

**Figure 2 cells-14-00799-f002:**
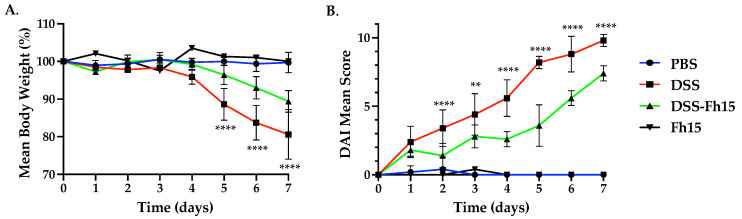
Fh15 treatment significantly reduces disease activity index and body weight loss in ulcerative colitis mice. (**A**) Daily body weight percentage of C57BL/6 male mice with ulcerative colitis treated with Fh15 (2 mg/kg body weight) on days 1, 3, and 5. (**B**) Daily disease activity index. Statistical significance between groups (*n* = 5) was determined by using two-way ANOVA with Dunnett’s multiple comparisons test, using the DSS group as the reference. **** *p* < 0.0001, ** *p* < 0.01.

**Figure 3 cells-14-00799-f003:**
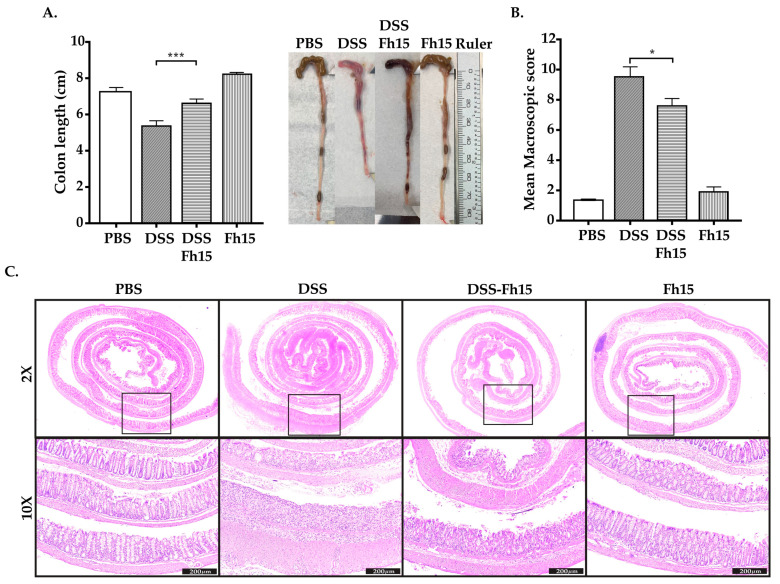
Fh15 significantly prevents colon shortening and reduces macroscopic score with notable decrease in histological alterations in male mice with DSS-induced ulcerative colitis. (**A**) Male colon length after three doses of Fh15 (2.0 mg/kg body weight). Colon measurement in centimeters was used as an indirect marker of inflammation. (**B**) Mean macroscopic score. (**C**) H&E staining of colon samples representing histological assessment. Top panel shows representative images at 2× objective. Bottom panel shows regions into the square magnified with a 10× objective and a scale bar of 200 μm. Statistical significance between groups (*n* = 5) was assessed by using one-way ANOVA with Dunnett’s multiple comparisons using DSS as the reference group. *** *p* < 0.001, * *p* < 0.05.

**Figure 4 cells-14-00799-f004:**
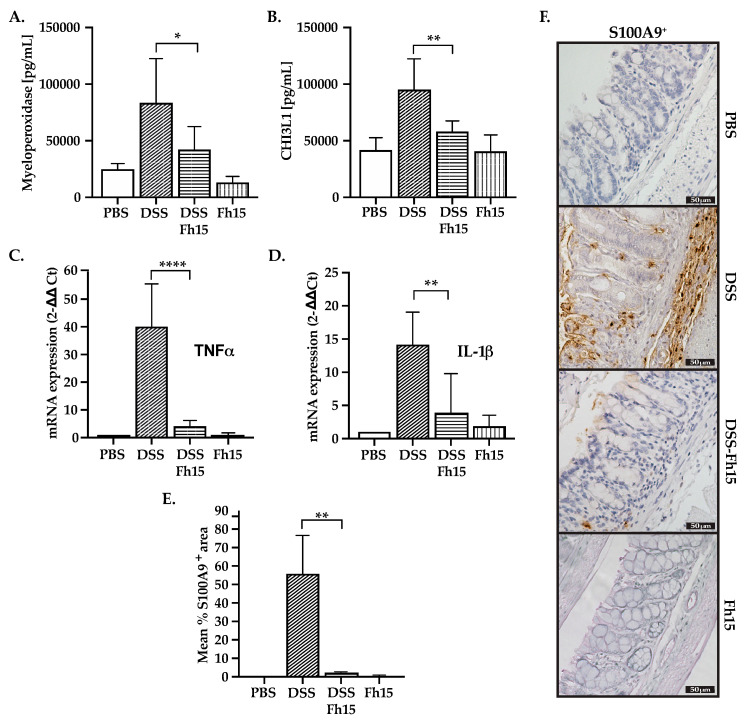
Fh15 reduces pro-inflammatory markers in male mice with DSS-induced ulcerative colitis. (**A**) Serum concentrations of myeloperoxidase and (**B**) chitinase-3-like protein measured in picograms per milliliter (pg/mL) using ELISA. mRNA expression levels of (**C**) TNFα and (**D**) IL-1β in the distal colon assessed via RT-PCR, with GAPDH used as housekeeping gene for internal reference. (**E**) The mean percentage of S100A9+ area in colonic samples and (**F**) representative immunostaining images of S100A9 observed with a 40× objective are shown. Statistically significant differences between groups (n = 5) were determined by using one-way ANOVA with Dunnett’s multiple comparisons, using DSS as the reference group. **** *p* < 0.0001, ** *p* < 0.01, * *p* < 0.05.

**Figure 5 cells-14-00799-f005:**
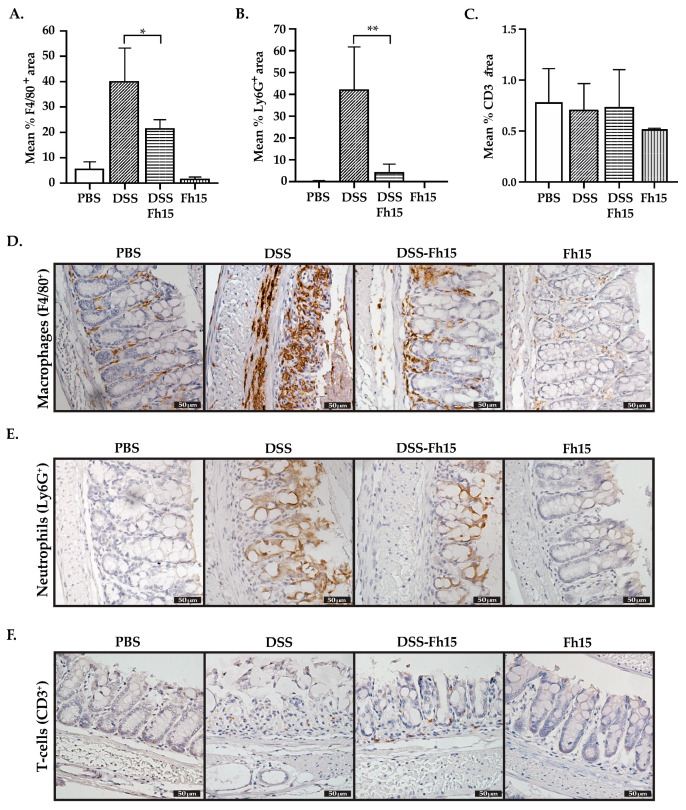
Fh15 effectively decreases macrophages and neutrophil populations in male mice with DSS-induced ulcerative colitis. (**A**) Mean percentage of F4/80+ area, (**B**) Ly6G+ area, and (**C**) CD3+ area in colonic samples. (**D**) Representative immunostaining images of macrophages population (F4/80+), (**E**) neutrophils population (Ly6G+), and (**F**) T-cells population (CD3+) observed with a 40× objective. Statistical analysis between groups (*n* = 5) were determined by using one-way ANOVA with Dunnett’s multiple comparisons, using DSS as the reference group. ** *p* < 0.01, * *p* < 0.05.

**Figure 6 cells-14-00799-f006:**
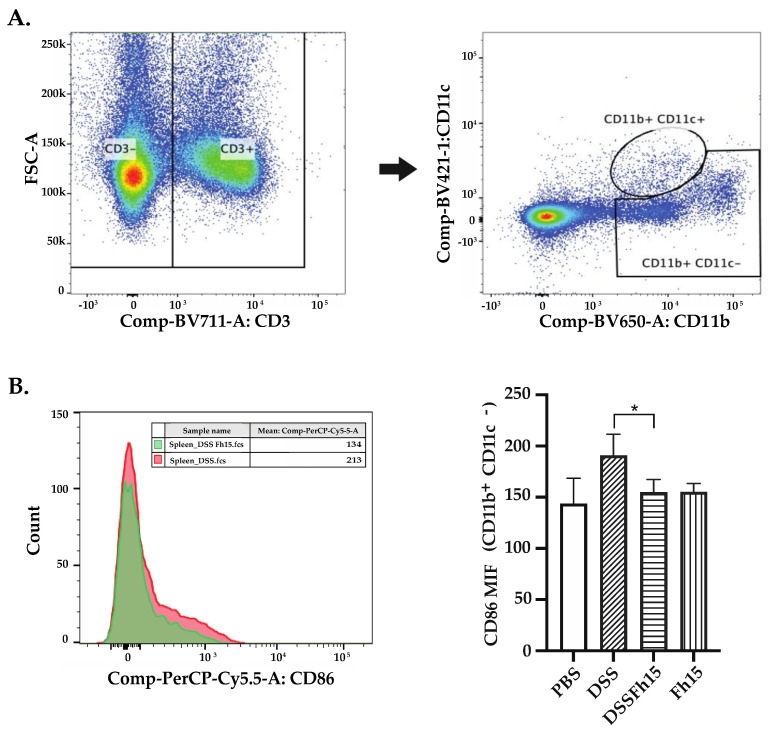
Fh15 effectively reduces CD86 expression on spleen CD11b+ CD11c− cell population. (**A**) Flow cytometry gating strategy used to identify CD3− cells from total splenocytes, followed by selection of CD11b+ CD11c− leucocyte cells. Intense red color indicates higher number of cells while green to blue shades represent lower cells account. (**B**) Within the CD11b+ CD11c− cell population, CD86+ cells were identified and quantified. The overlaid histogram illustrate the CD86 expression in CD11b+ CD11c− cell population, comparing samples from the DSS group (red) and the Fh15DSS group (green). The accompanying bar graph display the mean fluorescent intensity (MFI) of CD86 revealing a significant reduction in CD86 expression in the DSSFh15 group compared with the DSS-only group (*p* = 0.0161). Flow cytometry data were analyzed using FlowJo software (version 10.6.2) and statistical differences between groups (*n* = 5) were determined using one-way ANOVA followed by Dunnett’s multiple comparisons test. * *p* < 0.05.

**Table 1 cells-14-00799-t001:** Assessment of colon macroscopic score on day 7 after euthanasia, modified from Storr, M. et al., 2009 [24].

Macroscopic Damage	Score
Colon length	≥6 cm = 0 pt, <6 cm = 1 pt, or <5 cm = 2 pt
Inflamed length	Measurement in centimeters
Bowel thickness	Measurement in millimeters
Adhesion	0 = No Adhesion, 1 = Mild, 2 = Moderate, or 3 = Severe
Hemorrhage	Present = 1 or Absent = 0
Fecal blood	Present = 1 or Absent = 0
Diarrhea	Present = 1 or Absent = 0

**Table 2 cells-14-00799-t002:** RT-qPCR primers.

Gene	Primers	Sequence 5′-3′
GAPDH	Forward	CATGGCCTTCCGTGTTCCTA
	Reverse	CCTGCTTCACCACCTTCTTGAT
TNF-α	Forward	AAGCCTGTAGCCCACGTCGTA
	Reverse	AGGTACAACCCATCGGCTGG
IL-1β	Forward	GAAATGCCACCTTTTGACAGTG
	Reverse	TGGATGCTCTCATCAGGACAG

**Table 3 cells-14-00799-t003:** Primary antibodies (IgGs) used for chromogenic immunohistochemistry (IHC).

Marker	Target	Host Species	IHC Dilution	Cat. No.	Vendor
CD3ε F4/80	T-cells Macrophages	Rabbit Rabbit	1:100 1:100	ab16669 70076S	Abcam CST
LY6G	Neutrophils	Rabbit	1:75	87048S	CST
S100A9	Calcium binding protein A9	Rabbit	1:800	73425S	CST

## Data Availability

All data is presented in this manuscript and provided as Appendix A.

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
