# Peer review of "Fh15 Reduces Colonic Inflammation and Leukocyte Infiltration in a Dextran Sulfate Sodium-Induced Ulcerative Colitis Mouse Model"

_cells, 2025, doi:10.3390/cells14110799_

Round 1

Reviewer 1 Report (Previous Reviewer 1)

Comments and Suggestions for Authors

The authors of the paper have done an interesting job, addressing a perspective rarely addressed in terms of anti-inflammatory treatment. The reading has been enriching. I will share some comments below.

Minor comments

  • It would be interesting if the authors could add the number of animals used or experiments performed in the figure captions, together with the statistics (number of asterisks). This would further enhance the robustness of the system for the reader.

I really don't have much more comments. It is an interesting article. I would simply like to encourage the authors to go deeper into the molecular mechanisms behind these results, perhaps in the next paper.

Author Response

The authors of the paper have done an interesting job, addressing a perspective rarely addressed in terms of anti-inflammatory treatment. The reading has been enriching. I will share some comments below.

Minor comments

  • It would be interesting if the authors could add the number of animals used or experiments performed in the figure captions, together with the statistics (number of asterisks). This would further enhance the robustness of the system for the reader.

I really don't have much more comments. It is an interesting article. I would simply like to encourage the authors to go deeper into the molecular mechanisms behind these results, perhaps in the next paper.

Reply. Authors greatly appreciate your recommendations and we are happy our responses have addressed all your concerns and questions. As you requested, the number of animals was added to the legend of each figure. As you recommended, we are currently planning more studies aimed to explore the molecular mechanisms behind these results.

Reviewer 2 Report (Previous Reviewer 2)

Comments and Suggestions for Authors

The authors improved the image quality significantly except the colon morphology image.

Where are the colon morphology images of new experiment?

Author Response

The authors improved the image quality significantly except the colon morphology image.

Where are the colon morphology images of new experiment?

Reply. Authors greatly appreciate your recommendations and we are happy our responses to have been found satisfactory. The morphology images of the new experiment had been provided as supplementary figure. Therefore, we moved these data from the supplementary data file to the main text. Now, these images and analysis are shown in the Figure-5.

This manuscript is a resubmission of an earlier submission. The following is a list of the peer review reports and author responses from that submission.

Round 1

Reviewer 1 Report

Comments and Suggestions for Authors

This is a very interesting paper in which the authors use a recombinant parasite fatty acid binding protein for the treatment of chronic inflammatory diseases, in this case IBD. It is an interesting and different approach, used with other parasites, such as filariae, for the treatment of massive allergic syndromes. It has been a pleasant and stimulating read.

Could the authors confirm by rt-qPCR the different regenerative response genes in mice treated with Fh15 with respect to the other conditions? It would be interesting to have that information together with the genes related to the inflammatory response. Also, I would like to propose to the authors a second experiment in which they let the mice recover for a period of at least 7 days after DSS withdrawal, to check if there is a better repair in case of Fh15 administration, although I understand that I cannot include them in this study due to time issues.

I do not have many other comments. This is an interesting article and I think it should be considered for publication in Cells.

Author Response

Comment-1: Could the authors confirm by rt-qPCR the different regenerative response genes in mice treated with Fh15 with respect to the other conditions? It would be interesting to have that information together with the genes related to the inflammatory response. Also, I would like to propose to the authors a second experiment in which they let the mice recover for a period of at least 7 days after DSS withdrawal, to check if there is a better repair in case of Fh15 administration, although I understand that I cannot include them in this study due to time issues.

Response. We agree with this valuable comment. However, the comparison between the regenerative response genes in mice treated with Fh15 with respect to the other conditions is by itself a time-consuming approach that is out of the scope of the present study. The same could be said regarding the second experiment you propose of letting the mice recover for a period of 7 days after DSS withdrawal and checking if there is a better repair in case of Fh15 administration. We greatly acknowledge these two suggestions, which were immediately incorporated as part of the experimental design of incoming projects that we are designing. At the end of the discussion, we have added a detailed paragraph mentioning possible mechanisms of action for Fh15 and proposing further studies in which the putative role of Fh15 in the repairing of mucosa would be investigated.

Reviewer 2 Report

Comments and Suggestions for Authors

Author Response

Comment-1: In this research article, Figueroa-Gispert et. al. presents a detailed exploration of Fh15, a recombinant fatty acid-binding protein from Fasciola hepatica, as a potential therapeutic for ulcerative colitis (UC) using a dextran sulfate sodium (DSS)-induced mouse model. The study finding is well written and highlights Fh15's immunomodulatory properties, particularly its ability to reduce inflammation and leukocyte infiltration in the colon, providing evidence for its therapeutic potential in UC. Although the authors explored the effect of Fh15 on macrophages and neutrophils nicely, in DSS-induced colitis (an experimental model of UC), dendritic cells (DCs) and T cells are key players in promoting inflammation and immune dysregulation. The present study lacks any data regarding DC of T cells; especially regulatory T cells (Tregs). To challenge the therapeutic of Fh15, T-cell and B-cell deficient Rag1-/- mutant mouse model is a valuable tool to explore and strengthen its efficacy in clinical trial later.

Response: We greatly appreciate this comment. We agree in that it would be valuable to explore the role of Fh15 either on the DCs or T-reg cells. However, because these experiments are costly and time-consuming they were out of the scope of the present study. However, we greatly acknowledge these suggestions, which will be immediately incorporated as part of the experimental design of incoming projects that we are being designing. At the end of the discussion, we have added a paragraph mentioning the possible mechanism of action for Fh15 including the exploring of Fh15 effect on DC and Treg.

Major concerns:

  1. In most of the Figs (Fig 2 – Fig 5), the image quality is very poor; can be improved significantly by increasing font size, bar thickness etc.

Response. The image quality of these figures was improved.

  1. In Fig 3, colon images are not of publication grade, in Fig 3C., although it has been said 40X objective Lense was used, but it seems that it has been used 10X objective. It should be shown the whole colonic section (low objective 4X) along with partial (blown up, with higher objective 20X or 40X). The images are not of publication quality.

Response. We have made our best to improve the quality of these images. In Figure-3 we included a panel of images taken with the 10X objective and another showing the same images at higher magnification (40X).

  1. In Fig 4C, H&E images can be improved significantly including bar graphs.

Response. Fig. 4C does not show H&E images but HIC images showing the expression of S100A9. These images including bar graphs were improved.

  1. In Fig 5, H&E images can be improved significantly including bar graphs

Response. Fig. 5 does not show H&E images but HIC images showing the macrophages and neutrophils infiltration. These images including bar graphs were improved.

Round 2

Reviewer 2 Report

Comments and Suggestions for Authors

Thanks, the author improved the image qualities, though it could have been improved more. Isolation of T-cells, or any immune cells is not time consuming or expensive at all now a days. Making single cells from mouse colon, isolating immune cells by choice of maker antibody, followed by magnetic beads separation, then scanty mRNA can be used for making cDNA.   

Author Response

Response: We agree with this comment. The isolation of T-cells or any immune cells is not a time-consuming process now a days, however, what we should say is that we had not enough tissues for this purpose because we had used almost all tissues. Most sections had been sent to the immunopathology laboratory for IHC analysis and others had been used for RNA extraction, therefore not many tissues left. However, we understood that if we performed these additional experiments our manuscript would improve significantly and decided to do it. For this reason, our resubmission took longer than expected. We got more animals and replicate the experiment to obtain fresh tissues. Firstly, we performed an IHC analysis labelling colon tissue with specific antibody anti-CD3. We don’t find many T-cells in the colon of DSS-mice, which suggested that likely T-cells could not be required for inducing colitis in the DSS-model. When we revised the literature, we found that our assumption was correct. Previous studies had demonstrated that T cells are not required for induction of colitis in the DSS-model (Axelsson LG et al 1996, and Dieleman LA et al 1994) and other authors had later demonstrated that in this model T-cells play a role during the late acute phase and during chronic phase (Dieleman LA et al 1998, Melgar S et al. 2004, Hall LJ et al 2011). Therefore, we reasoned that instead to perform RT-PCR would be much more informative to explore what happened with the T-cell population in spleen that is the largest lymphoid organ. Spleen lymphocytes were labelled with a panel of specific antibodies for T-cells (CD3+, CD4+ and CD8+) and then analyzed by flow cytometry. Results confirmed the IHC results since no differences were observed between the T-cell populations of healthy mice and DSS-colitis or those treated with Fh15, which confirmed our initial findings with IHC. However, a leukocyte population classified as CD11b+CD11c- CD86+ was found significantly increased in DSS-colitis mice compared to healthy controls (p=0.0019). This leukocyte population was significantly suppressed by the Fh15-treatment (p=0.0161) (Figure 6), which suggest that Fh15 suppress the capacity of these cells to activate T cells. Since CD11b+ CD11c- exclude dendritic cells, we assume that this spleen leukocyte population could comprise monocytes, macrophages, neutrophils and natural killer cells, which are CD11b+. However, since CD86+ a co-stimulatory marker expressed on antigen-presenting cells that play a role in the T-cell activation, we speculated that Fh15 suppress this cell population during acute colitis in the DSS-model as mechanism to prevent the overactivation of T-cells in late acute and chronic phase of colitis. All these results were added and discussed in the manuscript. Moreover, we also tried to improve much more the quality of the IHC images showed in the figure-4 and figure-5 and provided as supplementary figures the original images of IHC showing all markers studied at two different magnifications (10X and 40X).
